# Generation of Monthly Precipitation Climatologies for Costa Rica Using Irregular Rain-Gauge Observational Networks

**Maikel Mendez [1,\*], Luis-Alexander Calvo-Valverde [2], Ben Maathuis [3] and Luis-Fernando Alvarado-Gamboa [4]**

[1] Instituto Tecnológico de Costa Rica, Escuela de Ingeniería en Construcción, Cartago 159-7050, Costa Rica
[2] Instituto Tecnológico de Costa Rica, Escuela de Ingeniería en Computación, Cartago 159-7050, Costa Rica; lcalvo@tec.ac.cr
[3] Department of Water Resources, Faculty of Geo-Information Science and Earth Observation (UT-I-ITC-WCC), University of Twente, Enschede Hengelosestraat 99 7514 AE, The Netherlands; b.h.p.maathuis@utwente.nl
[4] Departamento de Climatología e Investigaciones Aplicadas, Instituto Meteorológico Nacional, San José 5383-1000, Costa Rica; luis@imn.ac.cr
[\*] Correspondence: mamendez@tec.ac.cr; Tel.: +506-2550-2425

**Abstract:** Precipitation climatologies for the period 1961–1990 were generated for all climatic regions of Costa Rica using an irregular rain-gauge observational network comprised by 416 rain-gauge stations. Two sub-networks were defined: a high temporal resolution sub-network (HTR), including stations having at least 20 years of continuous records during the study period (157 in total); and a high spatial resolution sub-network (HSR), which includes all HTR-stations plus those stations with less than 20 years of continuous records (416 in total). Results from the kriging variance reduction efficiency (KRE) objective function between the two sub-networks, show that ordinary kriging (OK) is unable to fully explain the spatio-temporal variability of precipitation within most climatic regions if only stations from the HTR sub-network are used. Results also suggests that in most cases, it is beneficial to increase the density of the rain-gauge observational network at the expense of temporal fidelity, by including more stations even though their records may not represent the same time step. Thereafter, precipitation climatologies were generated using seven deterministic (IDW, TS2, TS2PARA, TS2LINEAR, TPS, MQS and NN) and two geostatistical (OK and KED) interpolation methods. Performance of the various interpolation methods was evaluated using cross validation technique, selecting the mean absolute error (MAE) and the root-mean square error (RMSE) as agreement metrics. Results suggest that IDW is marginally superior to OK and KED for most climatic regions. The remaining deterministic methods however, considerably deviate from IDW, which suggests that these methods are incapable of properly capturing the true-nature of spatial precipitation patterns over the considered climatic regions. The final generated IDW climatology was then validated against the Global Precipitation Climatology Centre (GPCC), Climate Research Unit (CRU) and WorldClim datasets, in which overall spatial and temporal coherence is considered satisfactory, giving assurance about the use this new climatology in the development of local climate impact studies.

**Keywords:** climatology; interpolation; irregular; kriging; precipitation; rain-gauge network; variance

## 1. Introduction

Precipitation climatologies represent the foundation of climate research in a variety of sectors, including agriculture, forestry, water resources management, hydrological and hydrogeological modelling, urban planning, flood inundation and biodiversity management; all of which need to adapt to future Climate Change impacts [1–3]. Precipitation climatologies have extensively been used to evaluate general circulation models (GCMs) and regional climate models (RCMs) by comparing their outputs to observational data sets [4,5]. The generation of precipitation climatologies is generally based on data coming from rain-gauge observational networks, which are recurrently sparse and unevenly distributed in many regions of the World [6]. Furthermore, historical records observed through these networks are frequently incomplete because of missing data during the observed period or insufficient number of stations in the study region [7,8].

The absence of homogeneous and continuous historical records can considerably limit the scope of local climate impact studies over specific areas of interest [9]. The reasons for such discontinuities include relocation of the observation stations, equipment and data-acquisition failure, cost of maintenance and changes in the surrounding environment due to anthropogenic or natural causes [10–12], which has motivated climatologists to re-examine historical weather-station records.

Rain-gauge observational networks are constructed with the intention of providing measurements that adequately characterize most of the non-trivial spatial variations of precipitation [13,14]. This is hard to accomplish if only continuous and regular observational networks are included in the generation of such climatologies. The situation becomes more severe in tropical regions, due to high spatial and temporal variability and scarce data availability [15].

On that premise, some climatologists may choose to favour a high spatial resolution (HSR) network by maximizing the amount of temporally discontinuous, irregularly distributed rain-gauges at the expense of reducing the extent to which the station records are lengthy and temporally commensurate [16]. On the contrary, a different group of climatologists may favour a high temporal fidelity by promoting a high temporal resolution network (HTR) at the expense of reducing the number of rain-gauges and, therefore, their ability to spatially resolve the climatic field of interest [17,18]. This is done by selecting records that are sufficiently long and temporally commensurate within a regularly distributed, temporally continuous network.

To this end, various attempts have been made to develop and improve global and regional gridded precipitation datasets, but they usually suffer from the lack of sufficient rain-gauge observational densities [19–21]. Nonetheless, historical records coming from rain-gauge observational networks even when discontinuous and irregular, represent the most reliable source of climatic information prior to the emergence of remote sensing products, and therefore continue to comprise the bases of most credible estimates for generating long-term time series of areal precipitation [6,22].

The reconstruction of precipitation climatologies largely relies on the application of spatial interpolation methods on data recorded by such observational networks [23]. Spatial interpolation is achieved by estimating a regionalized value at unsampled points from weights of observed regionalized variables. Interpolation methods can broadly be classified as either deterministic or geostatistical [24]. The fundamental principle behind deterministic methods is that the relative weight of an observed value decreases as the distance from the prediction location increases. Geostatistical methods however, are based on the theory of regionalized variables, and provide a set of statistical tools for incorporating the spatial correlation of observations in the data processing. Some authors have criticized deterministic methods since they are unable of accounting for temporal and spatial disaggregation [4]. Geostatistical methods, in contrast, have the capability of incorporating covariants fields, which are in principle denser than point field observations [25]. The relevant secondary information supplied by covariants can potentially improve interpolation results as long as they exhibit a strong correlation with the interpolated field [26]. Among such covariants, elevation has commonly been used [27,28].

Different interpolation methods lead to different errors depending on the realism of the assumptions on which they rely [29]. Geostatistical methods rely on the assumption of data stationarity, which requires normal distribution and homogeneous variance of the field variable [30]. Precipitation is a highly skewed, heteroscedastic and intermittent field in nature and therefore frequently contradicts the assumptions of data normality. This forces data transformation using analytical or numerical techniques, which cannot always satisfy the assumptions on normality and homoscedasticity [31]. Geostatistical interpolation methods nonetheless, have broadly been applied in the design, evaluation and monitoring of rain-gauge observational networks. Among such methods, kriging is one of the most popular geostatistical interpolation techniques—widely accepted due to its relatively low computational cost and its flexibility regarding input and output data [14]. When using kriging, several outputs can be generated besides the prediction field; this includes the estimation of the residual errors and the kriging variance. Kriging variance is also estimated on points where no observations exist and, consequently, provides a spatial view on the measure of performance [32]. Various methods of rain-gauge network performance have been developed using kriging interpolation techniques. For such methods, known as variance-reduction techniques, the performance evaluation of a rain-gauge network focuses on reducing the error variance of the average field value over a certain domain [33–35]. Most of these techniques take into account the number and location of rain-gauges to yield greater accuracy of areal precipitation estimation with minimum cost. Bastin et al. [33] used the kriging variance as a tool in rain-gauge network-design for optimal estimation of the areal average precipitation. The authors developed an iterative screening procedure that selected rain-gauges associated with the minimum kriging variance. Consequently, all available rain-gauges are prioritized and this information is used for adding or deleting rain-gauges within the network. Similarly, Kassim and Kottegoda [34] prioritized rain-gauges with respect to their contribution in kriging variance reduction in a rain-gauge network through comparative kriging methods. By contrast, Chebi et al. [35] developed a robust optimization algorithm blending kriging variance reduction and simulated-annealing to expand an existing rain-gauge network by considering Inverse Distance Weighting (IDF) curve-parameters evolution, which ultimately allowed the authors to optimally locate new rain-gauges in imaginary locations.

Consequently, the concept of kriging variance-reduction could also be used to determine whether a HSR rain-gauge network could yield a more accurate estimate of a point precipitation average than a HTR rain-gauge network for the same time step and region. If by using a HSR rain-gauge network, which is most likely temporally discontinuous and irregularly distributed over a certain domain for a specific time step, the resulting average interpolated precipitation yields a lower kriging variance, it is probably wiser to increase the density of the rain-gauge network at the expense of temporal fidelity by including more stations, even though their records may not exactly represent the same time step. This is of the utmost importance for the reconstruction of precipitation climatologies in many regions of the world, where rain-gauge network-densities, as well as their temporal resolution, are far from optimal [13]. As stated in the regional projection of the Fourth Assessment Report of the Intergovernmental Panel on Climate Change [36], Central America is one of the regions in the world that most likely will experience an important decrease in mean annual precipitation in the following decades. This trend has already been identified by previous Climate Change regional studies focusing on the Central American corridor [36–38]. In consequence, there is a heightened need to generate quality precipitation climatologies for such regions, even on the grounds of data-sparse discontinuous and irregular rain-gauge observational networks.

This research focuses on Costa Rica, Central America, where most of the territory has experienced some degree of precipitation change during the period 1961–1990, showing increases on the north-western Caribbean side and decreases on the Pacific side [39,40]. Accordingly, the objective of this study is to determine whether a high spatial resolution (HSR) rain-gauge network yields a more accurate estimate of average precipitation than a high temporal resolution (HTR) rain-gauge network, based on the application of kriging variance-reduction techniques. The performance of

various interpolation methods in the generation of monthly precipitation climatologies for the period 1961–1990 is then evaluated for all climatic regions of Costa Rica. Finally, the generated climatologies are validated against publicly available global precipitation datasets, including those from the Global Precipitation Climatology Centre (GPCC), the Climate Research Unit (CRU) and WorldClim.

## 2. Materials and Methods

### 2.1. Study Area

Costa Rica is located across the Central American isthmus between Panama and Nicaragua at almost its narrowest point (Figure 1b). The country is bordered by the Caribbean Sea to the east and the Pacific Ocean to the west, which favours oceanic and climatological influences from both oceans. Costa Rica occupies an area of 51,060 km$^2$ and is meridionally divided by northwest-southeast trending cordilleras of different topographic complexity which rise to over 3400 m (Figure 1a).

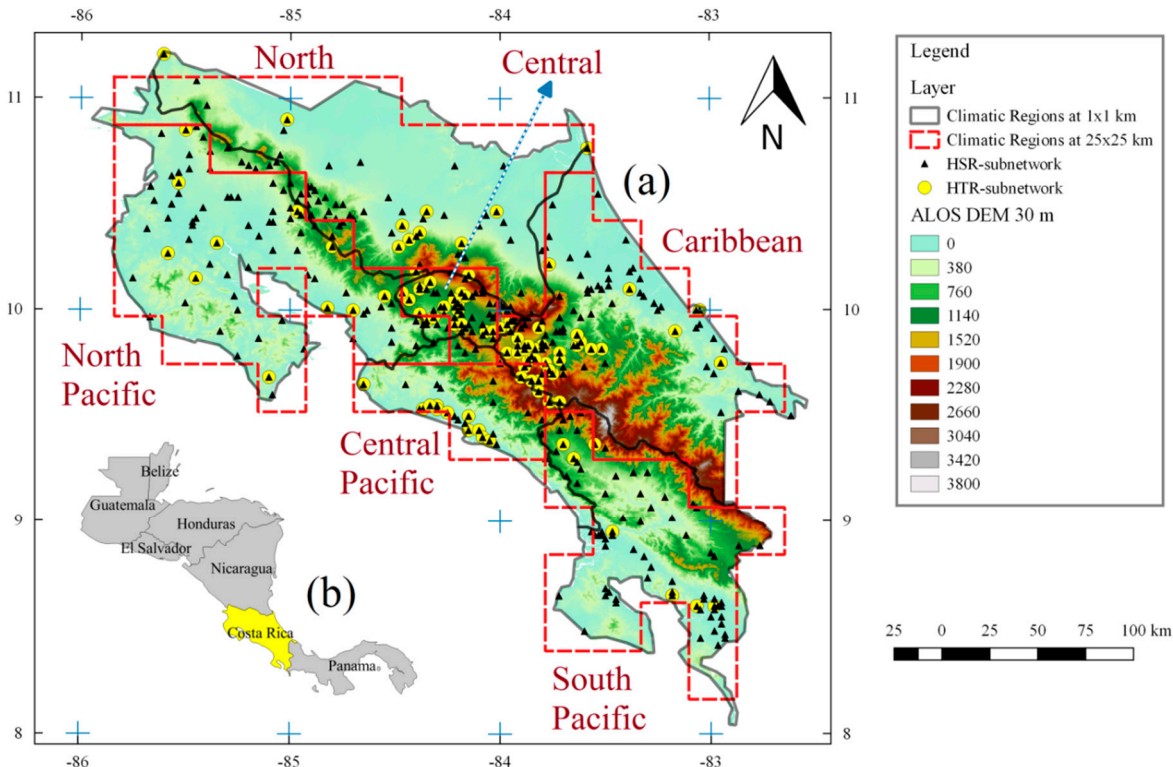

**Figure 1.** (**a**) Location of rain-gauge and Digital Elevation Model (DEM) for each climatic region in Costa Rica during the period 1961–1990. (**b**) Position of Costa Rica in Central America.

Coastlines and cordilleras however, do not run parallel to one another, thus displaying increased widths and elevations towards the south-east territory [41]. Precipitation variability in Costa Rica is driven by interactions between the local topography and a combination of the seasonal migration of the intertropical convergence zone (ITCZ), which includes sea breeze effects, monsoonal circulations, strong easterly trade winds, cold air masses from mid-latitudes in the winter and the perturbing influences of hurricanes and tropical cyclones in the Atlantic Ocean [42]. The complex pattern of precipitation regimes apparent in the country reflects influences at a variety of geographic scales. The temporal and spatial variability of precipitation in the country is heavily influenced by El Niño-Southern Oscillation (ENSO), which complex responses (warm or wet) vary in terms of their signs, magnitudes, duration and seasonality between those areas draining towards the Pacific and those draining towards the Caribbean. Consequently, mean monthly precipitation exhibits a strong seasonal cycle and regional variability [43], which can be observed by the monthly precipitation

derived from all available rain-gauge stations for the period 1961–1990 (Figure 2). Accordingly, the Instituto Meteorológico de Costa Rica (IMN) [44] has divided the Costa Rican territory into six separate climatic regions: North, Caribbean, North-Pacific, Central-Valley, Central-Pacific and South-Pacific, in which northeastern and southwestern domains are determined by the position and elevation of the aforementioned cordilleras (Figure 1a). These established regions reflect the separation that elevation imposes between Caribbean (northeastern) and Pacific (southwestern) sources of moisture and also reveals the important role played by elevation in controlling local precipitation [45]. This enables important insights into local climates, characterized by diverse and heterogeneous land surfaces.

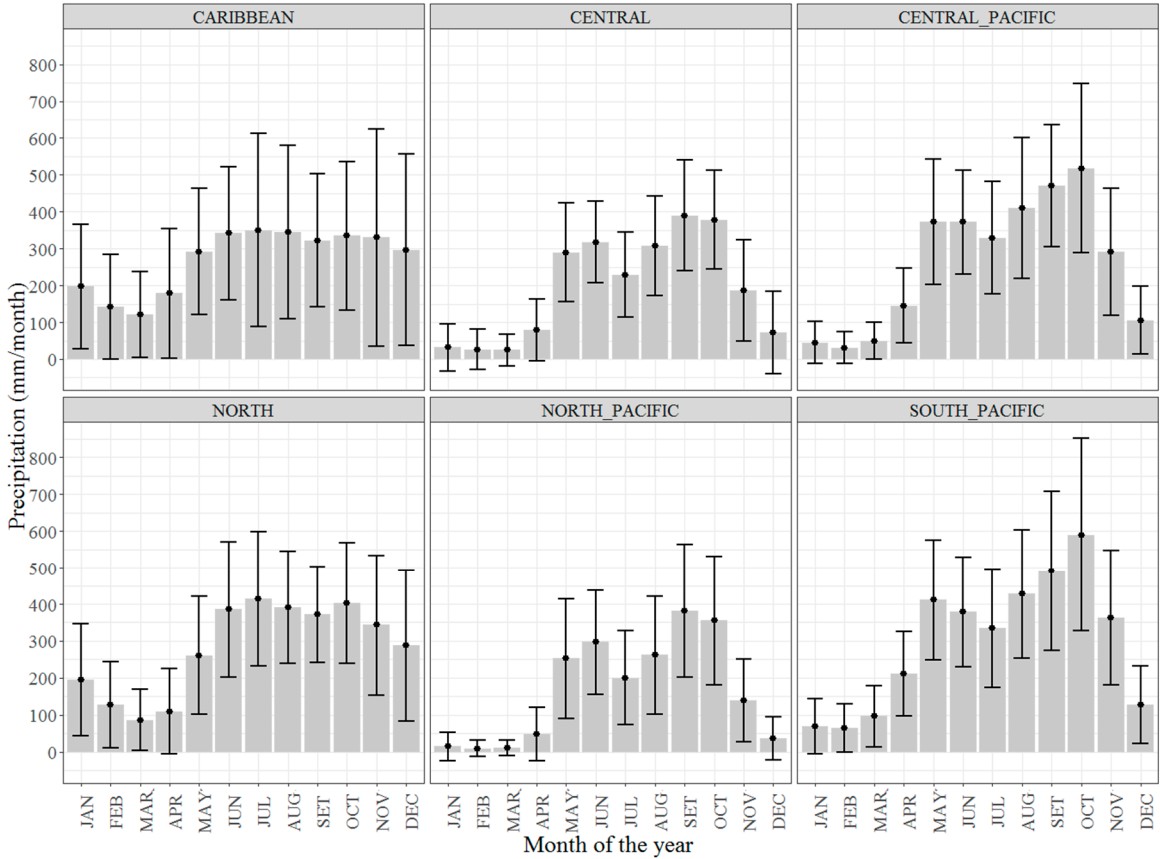

**Figure 2.** Mean monthly precipitation derived from all available rain-gauge stations for each climatic region in Costa Rica during the period 1961–1990. Error bars represent the standard deviation.

*2.2. Datasets and Data Transformation*

Aggregated monthly precipitation data were provided by Instituto Meteorológico of Costa Rica (IMN) for the period 1961–1990. A total of 416 rain-gauge stations were active during this period throughout the country, but their spatial distribution is irregular and not all stations registered continuously or records were not temporally commensurate. Consequently, two rain-gauge sub-networks were defined (Figure 1a): a high temporal resolution sub-network (HTR), which includes stations possessing at least 20 years of continuous records during the study period (157 in total); and a high spatial resolution sub-network (HSR), which includes all HTR-stations plus those stations with less than 20 years of continuous records (416 in total). Stations included in the HTR network had at least 90% of available monthly records during the 20-year defined temporal window. Therefore, the number of stations used at each time step varies over time within both sub-networks, with the HTR providing the most commensurate long-term monthly records.

An analysis of the temporal evolution of both sub-networks across all climatic regions shows that precipitation data were collected from a constantly changing configuration during the entire study

period, which peaks in the mid-1970s and starts to decrease in the mid-1980s, with a drastic drop after 1987, when many stations all around the country were either abandoned or relocated (Figure 3). The HSR sub-network satisfies World Meteorological Organization (WMO) standards of 250 km² / gauge for mountainous areas [46] in the Caribbean, Central-Valley and Central-Pacific regions during most of the 1970s and 1980s, with the Central-Pacific not reaching the minimum value during most of the 1960s. North, North-Pacific and South-Pacific regions, however, barely reach that standard during most of the 1970s and 1980s, with the North region being the most critically instrumented area, particularly during most of the 1960s. In contrast, the HTR sub-network only satisfies WMO standards for the Central-Valley region, since rain-gauge stations seemed to concentrate in the most populated region of the country. After 1987, however, only the Central-Valley region satisfies WMO standards. All other regions are way above the 250 km² / gauge after 1987 regardless of HSR or HTR, as there barely four operational stations in each region. Quality control of the IMN precipitation data was undertaken to identify possible systematic or acquisition errors and extreme outliers.

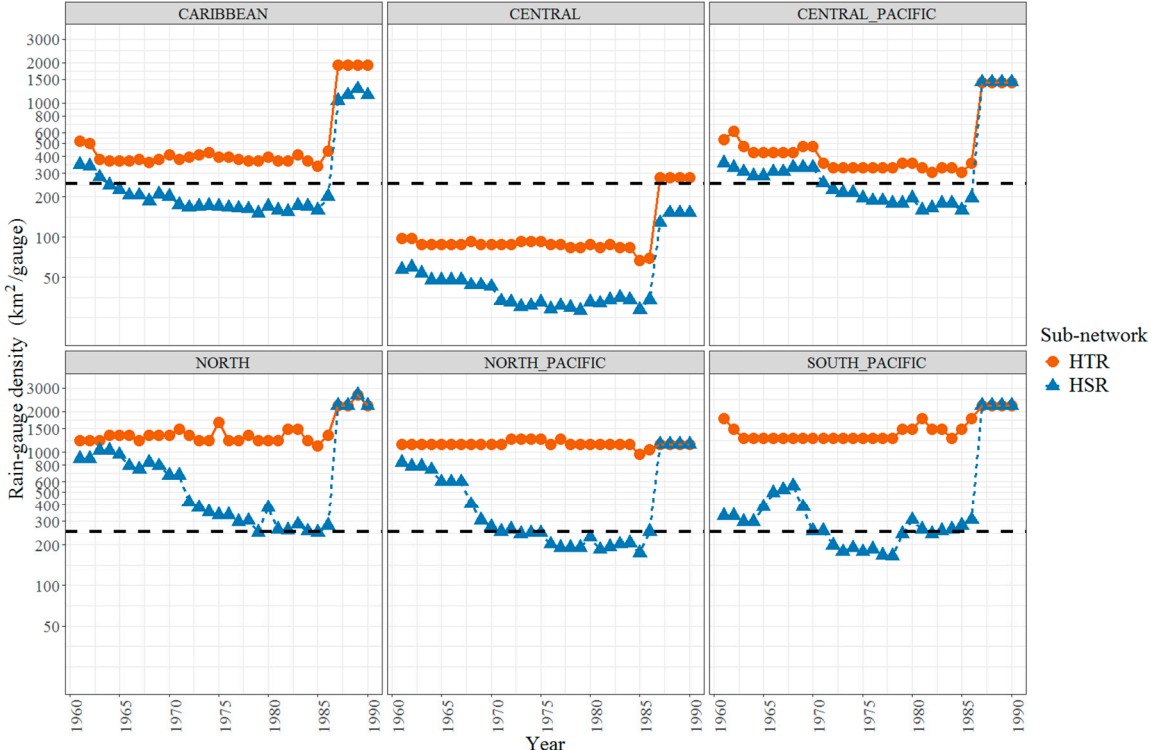

**Figure 3.** Temporal evolution of Instituto Meteorológico de Costa Rica (IMN) rain-gauge network density (km² / gauge) for each climatic region during the period 1961–1990. Black dashed-lines mark WMO density standards for mountainous areas (250 km² / gauge).

Since geostatistical interpolation methods rely on the assumption of data normality, records from both rain-gauge sub-networks were transformed using the Box–Cox optimization technique [47] to correct for non-Gaussianity and approximate normality. The transformation is dependent on the parameter Lambda only ($\lambda$).

$$Y^* = \begin{cases} \frac{Y^\lambda - 1}{\lambda} & \lambda \neq 0 \\ \log(Y) & \lambda = 0 \end{cases} \tag{1}$$

where $Y$ and $Y^*$ are the original and transformed variables, respectively; $\lambda$ is the transformation parameter.

Box–Cox transformation was individually applied to HSR and HTR datasets for each climatic region at a monthly basis, prior to the application of kriging interpolation. As suggested by Erdin et al. [48] and Woldemeskel et al. [49], possible values for "$\lambda$" were constrained to a minimum

value of 0.2 in order to avoid excessive data transformation. Precipitation estimates and kriging variance were subsequently back-transformed to their original units (mm/months). Logarithmic transformation was not considered because of the impossibility of transforming zero values. Kriging variance was also calculated using the raw non-transformed data for comparison purposes.

*2.3. Kriging Variance Reduction Efficiency*

Ordinary kriging (OK) is one of the most popular kriging estimators and can be described as a weighted average interpolation [30], where estimated values at target locations are calculated taking into account the distance of the neighbouring observed values to the location of the point to be estimated according to:

$$\hat{Z}(X_0) = \sum_{i=1}^{n} \lambda_i Z(X_i) \tag{2}$$

where $\hat{Z}$ is the estimated value at an unobserved location $X_0$, $Z$ is the observed value at the sampled location, $X_i$ and $\lambda_i$ are the kriging weights.

OK weights for sampled values are calculated based on the parameters of a variogram model, which provides the best linear unbiased estimator of point values with minimum error variance [15] according to:

$$\hat{\gamma}(h) = \frac{1}{2n} \sum_{i=1}^{n} [Z(X_i) - Z(X_i + h)]^2 \tag{3}$$

where $\hat{\gamma}$ is the semivariance as a function of distance, $h$ is the distance separating sampled points and $n$ is the number of pairs of sampled points.

A plot of $\hat{\gamma}(h)$ against $h$ is known as the observational variogram. Standard variograms models can then be fitted to the observations. OK weights are determined such as to minimize the estimation of the kriging variance:

$$Var[\hat{Z}(X_0)] = E\left[(\hat{Z}(X_0) - Z(X_0))^2\right] \tag{4}$$

where *Var* is the kriging variance and $Z(X_0)$ is the true value expected at point $X_0$.

OK weights are finally calculated by relating the semivariance $\hat{\gamma}$ to a system of linear equations known as the ordinary kriging system (OKS):

$$\begin{cases} \sum_{i=1}^{n} \lambda_i \hat{\gamma}(di_j) + \mu = \hat{\gamma}(di_0); \ for j = 1, \dots n \\ \sum_{i=1}^{n} \lambda_i = 1 \end{cases} \tag{5}$$

where $\hat{\gamma}(di_j)$ and $\hat{\gamma}(di_0)$ indicate the variogram values that come from the standard variogram models for the distance $di_j$ and $di_0$ respectively, $di_j$ is the separation distance between sampling points $X_i$ and $X_j$, $di_0$ is the separation distance between the sampling point $Xi$ and the target location, and $\mu$ is the Lagrange multiplier.

The robustness of OK significantly depends on the proper selection of standard variograms models that quantify the degree of spatial autocorrelation in the dataset [24]. Since the duration, magnitude and intensity of precipitation events vary across space and time, it is not realistic to adopt a unique variogram for all precipitation events, irrespective of seasonal and meteorological conditions [14]. Thus, selecting an appropriate model to capture the features of the data is critical [9]. Variogram fitting should reflect specific spatial structures of particular time lapses and therefore, the use of stationary variograms should be avoided. In this study, parameters values of the range, nugget and sill for standard variograms models (Spherical (Sph), Exponential (Exp), Gaussian (Gau), Matern (Mat) and Matern–Stein (Sten)) were automatically and individually fitted to HTR and HSR datasets at

each time step, such as to minimize the weighted sum of squares of differences between experimental and model variogram values:

$$WSS = \sum_{i=1}^{n} \omega(h_i)[\hat{\gamma}(h_i) - \gamma(h_i)]^2 \tag{6}$$

where $\gamma$ is the experimental semivariance, $\hat{\gamma}$ is the model semivariance, $h$ is the distance separating sampled points and $\omega$ is the relative weight assigned as a function of distance.

To assess whether the HSR sub-network yielded a lower kriging variance than the HTR sub-network for the same time step, the kriging variance reduction efficiency (KRE) function of the spatially-averaged kriging variance between the two sub-networks was employed for each of the six climatic regions, according to:

$$KRE = \left( \frac{Var_{HTR}\left[\hat{Z}(X_0)\right] - Var_{HSR}\left[\hat{Z}(X_0)\right]}{Var_{HTR}\left[\hat{Z}(X_0)\right]} \right) 100 \tag{7}$$

where *KRE* is the kriging variance reduction efficiency (%), $Var_{HTR}$ is the HTR spatially averaged kriging variance and $Var_{HSR}$ is the HSR spatially averaged kriging variance.

Positive *KRE* values for a certain climatic region and time step indicate that the HSR sub-network yields a more accurate estimate of a point precipitation average than the HTR sub-network. Negative values indicate the opposite. *KRE* was also calculated using the raw non-transformed data for comparison purposes.

### 2.4. Interpolation Methods and Experimental Setup

Deterministic and geostatistical interpolation methods (Table 1) were selected to produce spatially continuous precipitation climatologies for each climatic region of Costa Rica during the period 1961–1990 based on datasets from the HSR sub-network only (Figure 1a). All spatial interpolation and data processing was executed using the R programming language (v3.5.2) [50] along with specialized R packages. Geostatistical modelling, spatio-temporal data analysis and raster generation were implemented by combining functionalities of the gstat (v1.1.6), sp (v1.3.1), raster (v2.8.4), RSAGA (v1.3.0) and rgdal (v1.3.6) packages. Since the spatial structure of precipitation data varies in space and time, OK automatic variogram fitting analysis was conducted separately for each sub-network and climatic region at a monthly time step using the R packages automap (v1.0.14), which minimizes the weighted sum of squares of differences between experimental and model semivariogram (Equation (1)). All distances were calculated according to the official Costa-Rica Transverse-Mercator (CRTM05) projected-coordinate-system using a spatial grid resolution of $1 \times 1$ km, which was selected primarily on the grounds of computational costs. The selected interpolation methods were chosen on the basis of (a) previous use in meteorology and climatology applications [13,24,51–53]; (b) continuity of recorded data; (c) location and distribution of available rain-gauges and (d) computational cost. Topographic information was derived from the Advanced Land Observing Satellite (ALOS) AW3D-30 m (30) Digital Elevation Model (DEM), which was subsequently resampled to a $1 \times 1$ km spatial resolution using the bilinear resampling technique. The R-code, raw-data, results and ggplot2 graphic-code are presented in Supplementary Materials. As recommended by Daly et al. [26], resampled elevation from the DEM was preferred over the actual station elevations points to improve spatial representation and generalization of orographic and convective precipitation mechanisms.

**Table 1.** Selected interpolation methods and relevant R packages.

| Abbreviation | Method | R_package | Class |
| --- | --- | --- | --- |
| IDW | Inverse Distance Weighting | gstat, sp, raster | Deterministic |
| TS2 | Trend surface, 2nd. polyn. surface | gstat, sp, raster | Deterministic |
| TS2PARA | Trend surface, 2nd. parab. surface | gstat, sp, raster | Deterministic |
| TS2LINEAR | Trend surface 2nd planar surface | gstat, sp, raster | Deterministic |
| TPS | Thin Plate Spline | gstat, sp, raster, rsaga | Deterministic |
| MQS | Modified Quadratic Shepard | gstat, sp, raster, rsaga | Deterministic |
| NN | Nearest Neighbour | gstat, sp, raster, rsaga | Deterministic |
| OK | Ordinary Kriging | gstat, raster, automap | Geostatistical |
| KED | Kriging with External Drift | gstat, raster, automap | Geostatistical |

### 2.5. Performance Assessment of the Interpolation Methods

Performance of the various spatial interpolation methods was evaluated using a leave-one-out cross validation (LOOCV) technique on the HSR network only (Figure 1a). Cross validation statistics serve as diagnostic tools to determine whether the performance of the selected interpolation method was acceptable. In LOOCV, a subset of stations from the entire data set is temporarily removed and the values at the same locations are estimated using the remaining stations (in this case, 25% of the stations without repetitions). LOOCV was sampled randomly with no data repetition due to the temporal discontinuities of the HSR sub-network. The procedure was repeated until all the stations in the data set were temporarily removed in turn and estimated. Cross validation was limited to the period 1961–1987 for all climatic regions except for the Central-Valley and Caribbean regions, since after 1987 there were not sufficient stations to properly calculate semi-variograms for OK and KED. Two indicators were used to quantitatively compare interpolated estimates against rain-gauge observations, the mean absolute error (MAE) and the root-mean square error (RMSE) according to:

$$MAE = \frac{1}{n}\sum_{i=1}^{n}|P_i - O_i| \tag{8}$$

$$RMSE = \sqrt{\frac{1}{n}\sum_{i=1}^{n}(P_i - O_i)^2} \tag{9}$$

where $P_i$ and $O_i$ are the predicted and observed values respectively.

The *MAE* is an absolute measure of bias that varies between 0 to $+\infty$. A *MAE* value close to 0 indicates an unbiased prediction. The *RMSE* ranges from 0 to $+\infty$, and it is used for checking the estimation accuracy between observed and predicted values. A *RMSE* value close to 0 indicates a higher accuracy in estimation. The two indicators were calculated at each time step and presented as monthly average for each climatic region.

### 2.6. Validation Datasets

The Global Precipitation Climatology Centre (GPCC), the Climate Research Unit (CRU), and WorldClim global precipitation datasets were used to assess the performance of the generated IDW climatology for all climatic regions of Costa Rica based on data from the HSR network (Figure 1a). The full GPCC version 2018 dataset [21], which covers the period 1891–2016 at a 0.25° spatial resolution (~25 km), is the most accurate in situ centennial monthly global land-surface precipitation product of GPCC. It is based on the ~80,000 stations worldwide that feature record durations of 10 years or longer. The data coverage per month varies from ~6000 (before 1900) to more than 50,000 stations. Since GPCC covers the period 1891–2016, monthly totals were isolated for the period 1961–1990 only. The CRU-CL version 2.0 monthly precipitation dataset [19], which covers the period 1961–1990 with a spatial resolution of 10 min (~18.5 km), was constructed based on observations of a number of sources including national meteorological agencies and archive centres, the WMO and the International Centre

for Tropical Agriculture (CIAT). The CRU-CL includes not only precipitation but also temperature and relative humidity among other variables. The WorldClim version 1.0 monthly precipitation dataset [20], which was generated through interpolation of average monthly climate data from weather stations, covers the period 1960–1990 and was distributed at spatial resolutions of 30 arc-s, 10 min, 5 min and 2.5 min. Major climate databases used in the generation of WorldClim include the Global Historical Climatology Network (GHCN), the Food and Agriculture Organization of the United Nations (FAO), the WMO, the International Centre for Tropical Agriculture (CIAT) and a number of additional national databases. For comparison purposes, all global precipitation datasets, along with the final IDW-generated climatology for Costa Rica, were resampled to a 25 × 25 km spatial resolution using the bilinear resampling technique; mainly selected on the basis of the GPCC 0.25° (~25 km) original spatial resolution (Figure 1a). The resulting spatially averaged time series were analysed for each climatic region using the mean absolute error (MAE), the root-mean square error (RMSE) and the Pearson's correlation coefficient (CORR), intended to quantify the goodness of fit between the local precipitation climatology and the aforementioned global datasets.

## 3. Results and Discussion

### 3.1. Kriging Reduction Efficiency

Monthly box-plots of the kriging reduction efficiency (KRE) for the period 1961–1990 show significantly diverse responses among all climatic regions for both transformed and non-transformed datasets (Figure 4). Concerning transformed datasets, KRE returns predominantly positive values throughout most of the study period regardless of the climatic region, which proves that in most cases, the HSR sub-network yields lower kriging variance than the HTR sub-network for the same time step. As kriging variance greatly depends on parameters derived from standard-variograms (range, nugget and sill), which fitting reflects specific spatial structures of particular time steps; predominantly KRE positive values suggest that variograms derived from the HTR sub-network are insufficient to properly represent the regional variability of monthly precipitation. In consequence, the HSR sub-network captures a more detailed spatio-temporal distribution of the precipitation patterns over most climatic regions of Costa Rica, even when the network is temporally discontinuous and irregularly distributed. Despite the HTR sub-network providing the most commensurate, long-term monthly precipitation records, its density remains below WMO standards for mountainous areas (Figure 3), which is particularly vulnerable in the analysis and modelling of spatio-temporal climatic variability.

Specifically, the North, North-Pacific and South-Pacific regions show entirely positive values and similar KRE quantile distributions throughout the study period, with lower median values during the driest months (JFMA) and a tendency to increase as average precipitation also increases. KRE then stabilizes during the wettest months (June through October) with narrower quantile distributions, to finally decrease again at the end of November. This implies that KRE is more effective during the wet season than during the dry season for these regions, with the North-Pacific and South-Pacific regions draining towards the Pacific Ocean and the North region discharging towards the Caribbean Sea (Figure 1a). The Central-Pacific region nonetheless, even when draining directly towards the Pacific Ocean, exhibits a considerable distinct KRE behaviour when compared to the North-Pacific and South-Pacific regions, since quantile distributions spread over the entire range (−100% to 100%) indistinctly of dry or wet months (Figure 4).

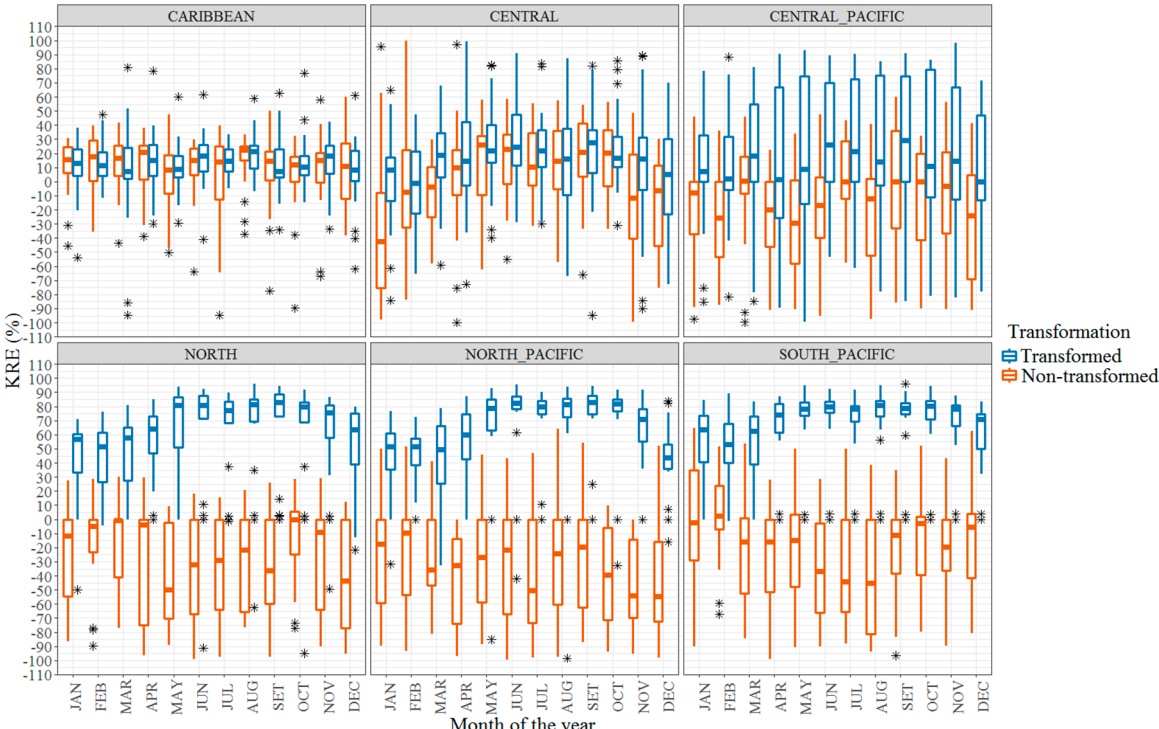

**Figure 4.** Kriging reduction efficiency (KRE) for transformed and non-transformed datasets for each climatic region during the period 1961–1990. Points marked with (*) symbol represent outliers.

During the months of April, May and December, which represent the transition between the dry and wet seasons, nearly 50% of the observations reach negative values as does October (the wettest month) for the Central-Pacific region. This region exhibits higher network densities than the North-Pacific or South-Pacific regions, but the HTR sub-network is highly concentrated along the Pacific coastline, with only a few stations in the mountainous areas (Figure 1a). This configuration suggests that during the analysed period (1961–1990), the Central-Pacific sub-network captured spatially localized precipitation events throughout the year, which complex responses vary in terms of their magnitudes, duration and seasonality. The occurrence of spatially localized precipitation events is exacerbated by various climatic effects associated to the intertropical convergence zone (ITCZ) [41,42], which most certainly occur at daily or hourly time scales and are ultimately aggregated into monthly timescales. Subsequently, HTR fitted variograms exhibit lower nugget, lower sill and shorter ranges values as compared to their HSR counterpart, which nevertheless is applicable to the entire Central-Pacific region. To illustrate such situation, the HTR and HSR fitted variograms for August 1971 are compared (Figure 5). In this case, the kriging-standard-error (square root of the kriging variance) for the HTR sub-network (expressed in mm/month) tends to represent only a small and localized fraction of the Central-Pacific coastline (Figure 5a), whereas the same metrics are more spatially distributed for the HSR sub-network (Figure 5b). In consequence, even when the average kriging variance for the HTR sub-network (0.112) is lower than its HSR counterpart (0.136), which results in a negative KRE value (−21.401%), the metrics seem to be spatially biased due to the abovementioned concentration of rain-gauge stations near the Pacific coastline. The presence of KRE negative values in the Central-Pacific region should not suggest that the HTR sub-networks yield a more accurate estimate of a point precipitation average than the HSR sub-networks do, since this tendency is highly variable and does not replicate during the entire period of analysis. In summary, OK is unable to explain a high portion of the spatial variability within the Central-Pacific region if only stations from the HTR sub-network are included. Similar arguments can be used to explain the presence of scattered KRE negative values in the Caribbean and Central-Valley regions, both of which have the highest and more temporally stable network densities of all climatic regions (Figure 3).

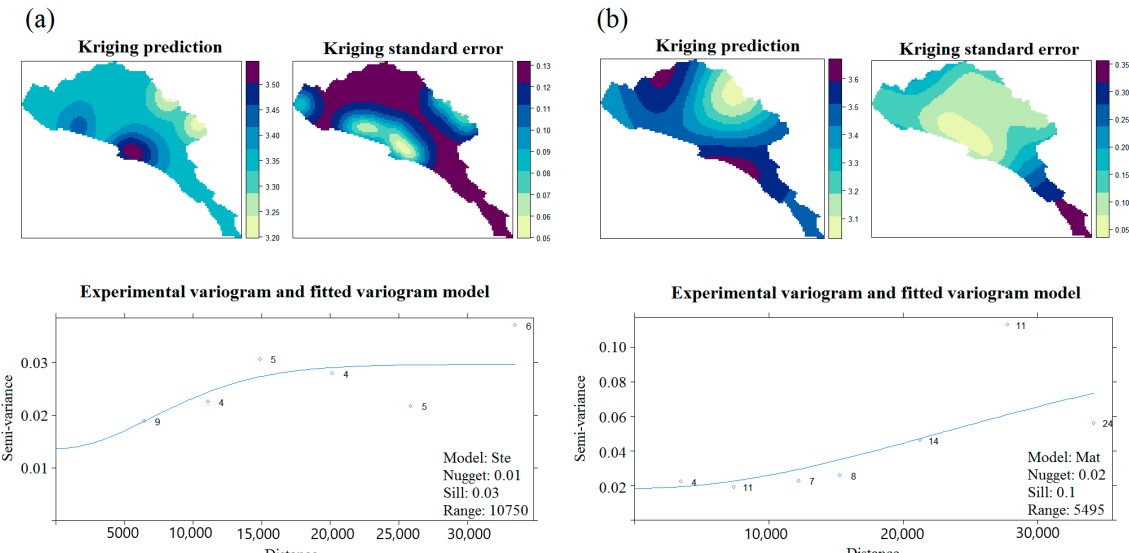

**Figure 5.** Kriging prediction, kriging-standard-error and fitted variogram for the HTR (**a**) and HSR (**b**) sub-networks for August 1971. Central-Pacific climatic region, Costa Rica.

In the case of the Central-Valley region, in agreement to the Central-Pacific region; the transitional months of April and December show the widest KRE quantile distributions, with a clear tendency to stabilize during the wettest months (May to November). Similar to the North, North-Pacific and South-Pacific regions, KRE is more effective during the wet season than during the dry season for the Central-Valley region. In this case, the incidence of higher network densities does not prevent the presence of KRE negative values during the driest months.

On the other hand, performance of the kriging variance-reduction (KRE) noticeably benefited from the Box–Cox optimization technique when applied to the North, North-Pacific, South-Pacific and Central-Pacific regions, since in most cases non-transformed datasets produced predominantly negative KRE values regardless of dry or wet months (Figure 4), which proves that data transformation did improve Gaussianity of the precipitation field. Furthermore, for the North, North-Pacific and South-Pacific regions, interquartile ranges are hardly distinguishable for most non-transformed boxplots, indicating that precipitation is a highly skewed, heteroscedastic and intermittent field in nature, which usually contradicts the assumptions of data normality [31]. This seems particularly evident in Costa Rica, as mean monthly precipitation exhibits a strong seasonal cycle and regional variability (Figure 2). Data transformation, nonetheless, is not as beneficial for the Caribbean and Central-Valley regions, both of which exhibit the highest network densities (Figure 3). On one hand, non-transformed KRE values outperform their corresponding transformed counterpart during the wettest months (May to October) for the Central-Valley region. The opposite situation occurs during the driest months. On the other hand, the Caribbean region, which exhibits the most persistent year-round precipitation regime (Figure 2), shows little gain in applying data transformation, suggesting a more normally distributed spatial pattern as compared to the remaining climatic regions. In summary, data transformation seems to be more effective in regions with lower network densities. In regions with higher network densities, however, data transformation is more effecting during the wettest months.

### 3.2. Temporal Evolution of the Observational Network

The temporal increase in rain-gauge density observed from 1961 to 1987 (Figure 3) does not seem to significantly impact KRE values for the North, North-Pacific, South-Pacific and Caribbean regions regardless of wet or dry season, since temporal observed-trends remain fairly stable throughout that segment of the study period (Figure 6).

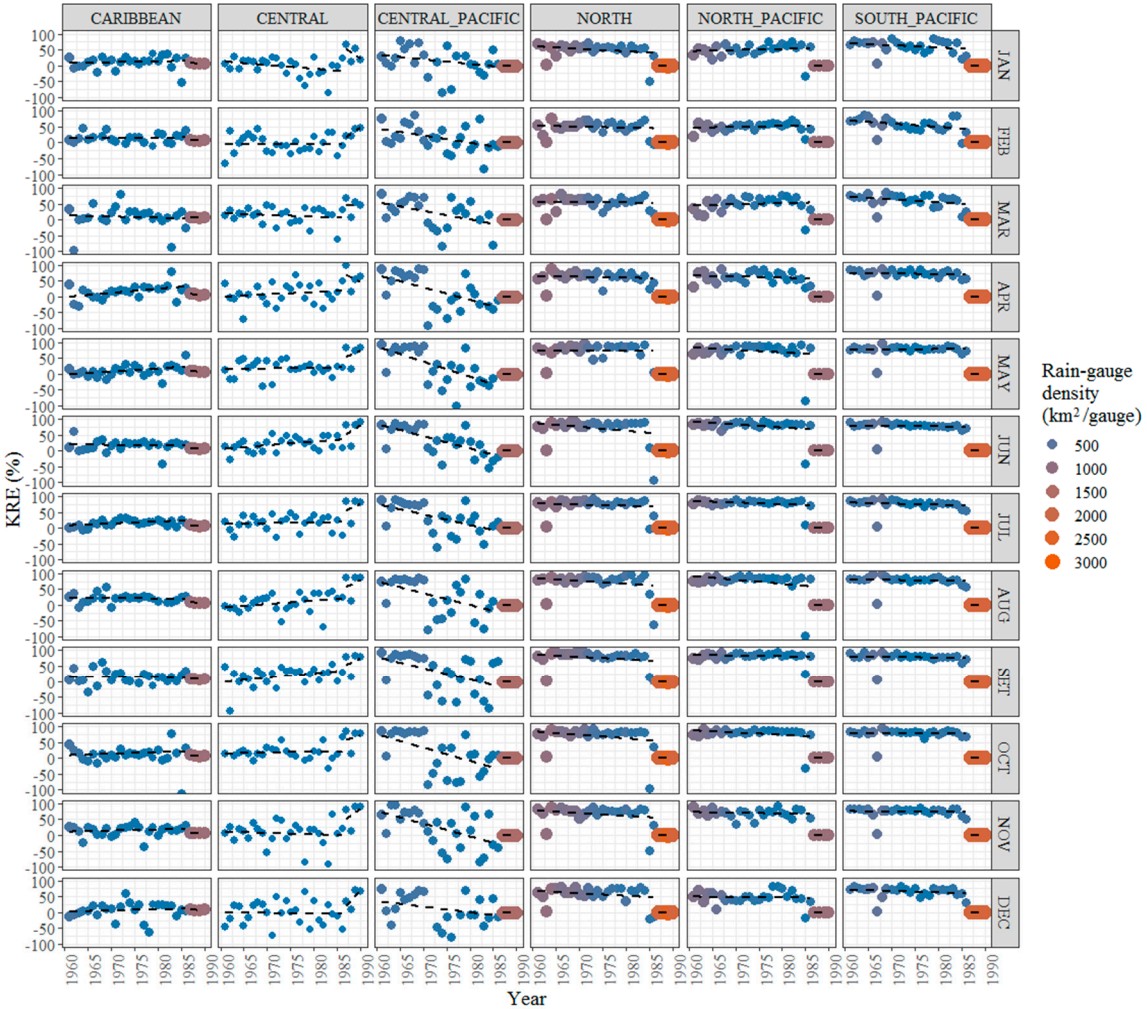

**Figure 6.** Kriging reduction efficiency (KRE) temporal evolution for the transformed HSR sub-network datasets for each climatic region during the period 1961–1990.

This implies that prior to 1987, the HSR rain-gauge sub-network was sufficiently dense to capture the spatio-temporal variability within each of these climatic regions in more detail. The Central-Pacific and Central-Valley regions nonetheless, do not follow the same pattern when compared to the remaining climatic regions. In the first case, there is in fact a temporal decrease in KSE as rain-gauge density increases for all months except June, July and December, which once again could be related to the highly concentrated number of stations along the Pacific coastline. The Central-Valley region, on the other hand, shows a contradictory behaviour with a decreasing KRE tendency during the driest months (JFM) and an increasing KRE tendency during the wettest months (ASO). In the case of the Central-Valley region, this could also be related to the occurrence of spatially localized precipitation events during the dry season that do not necessarily generate semi-variograms representative of the entire climatic region. Once again, mean monthly precipitation exhibits a strong seasonal variability all around the country [39,43].

After the abrupt drop in the number of rain-gauge stations experienced around 1987 (Figure 3), all climatic regions show densities way above 250 km²/gauge, which ultimately caused a drastic KRE decrease, irrespectively of wet or dry seasons. This is directly related to the lesser number of available rain-gauge stations at a national level, making both sub-networks (HTR and HSR) very much alike in proportion. As the number of rain-gauge stations between the two sub-networks reaches a constant value, the estimation of the KRE statistic becomes meaningless, since the ratio of spatially averaged kriging variances approaches zero. This is to be expected, since progressive improvement on the

accuracy of interpolated results with increasing rain-gauge densities have been found in various studies dealing with the application of kriging variance reduction techniques [28,54,55]. During the evaluation of an experimental catchment in South West England, Otieno et al. [28] found that rain-gauge density had an effect on the accuracy of interpolated results regardless of geostatistical or deterministic methods, since they found a gradual improvement in error statistics with a corresponding increase in the gauge density. In another study by Villarini et al. [54] over the Brue catchment in south-western England, the effect of both temporal resolution and gauge density on the performance of remotely sensed precipitation products was evaluated. Their results showed progressive improvement of interpolated products with increasing rain-gauge density.

In a similar study in Lower Saxony, Germany, Berndt et al. [55] intended to investigate the performance of merging radar and rain-gauge data for different temporal resolutions and rain-gauge network densities, comparing among other aspects the influence of temporal resolution and gauge density on variety of interpolation methods. Their findings indicate that any increase in sampling density could improve the prediction accuracy of the considered interpolation methods used in spatial prediction. The general trend observed at higher rain-gauge densities found by these authors also agree with the findings of Li et al. [30] and Yang et al. [1], which established that the accuracy of the methods used for spatial prediction increases as rain-gauge sample density also increases.

### 3.3. Performance of the Interpolation Methods

Cross validation heatmaps of MAE (Figure 7a) and RMSE (Figure 7b) mean monthly values (colour gradient in absolute units of mm/month, text-labels expressed as percentage with respect to the corresponding mean monthly precipitation) show that for all climatic regions, IDW, OK and KED rank the highest positions of all evaluated interpolation methods (Table 2), since significantly lower deviations (both absolute and percentage) are obtained when compared to the remaining interpolation methods. In the case of the Central-Pacific region, nonetheless, IDW, TPS and KED occupy the highest-ranking positions regarding MAE, and IDW, NN and KED regarding RMSE. All other deterministic methods considerably deviate from IDW, OK and KED both in absolute units and as percentage; particularly MQS, which seems unreliable to apply in all regions except the Caribbean and the Central-Valley. Furthermore, MAE and RMSE results from IDW, OK and KED methods reveal similar patterns of monthly precipitation distributions within each climatic region, whereas all remaining deterministic methods produced considerably different seasonal patterns throughout the year, mainly during the wettest months (July to November).

**Table 2.** Relative ranking of the various interpolation methods per climatic region.

| METHOD | CARIBBEAN | NORTH | NORTH_PACIFIC | SOUTH_PACIFIC | CENTRAL_PACIFIC | CENTRAL | OF |
|---|---|---|---|---|---|---|---|
| IDW3 | 1 | 2 | 1 | 1 | 1 | 1 | MAE |
| TS2 | 7 | 7 | 7 | 7 | 8 | 4 | MAE |
| TS2PARA | 6 | 6 | 8 | 6 | 7 | 6 | MAE |
| TS2LINEAR | 8 | 4 | 4 | 4 | 5 | 5 | MAE |
| NN | 4 | 5 | 5 | 5 | 4 | 8 | MAE |
| TPS | 5 | 8 | 6 | 8 | 2 | 7 | MAE |
| MQS | 9 | 9 | 9 | 9 | 9 | 9 | MAE |
| OK | 3 | 1 | 3 | 2 | 6 | 2 | MAE |
| KED | 2 | 3 | 2 | 3 | 3 | 3 | MAE |
| IDW3 | 1 | 2 | 1 | 1 | 1 | 1 | RMSE |
| TS2 | 7 | 7 | 7 | 7 | 8 | 4 | RMSE |
| TS2PARA | 6 | 6 | 8 | 6 | 7 | 6 | RMSE |
| TS2LINEAR | 8 | 4 | 4 | 4 | 5 | 5 | RMSE |
| NN | 4 | 5 | 6 | 5 | 2 | 8 | RMSE |
| TPS | 5 | 8 | 5 | 8 | 4 | 7 | RMSE |
| MQS | 9 | 9 | 9 | 9 | 9 | 9 | RMSE |
| OK | 3 | 1 | 3 | 2 | 6 | 2 | RMSE |
| KED | 2 | 3 | 2 | 3 | 3 | 3 | RMSE |

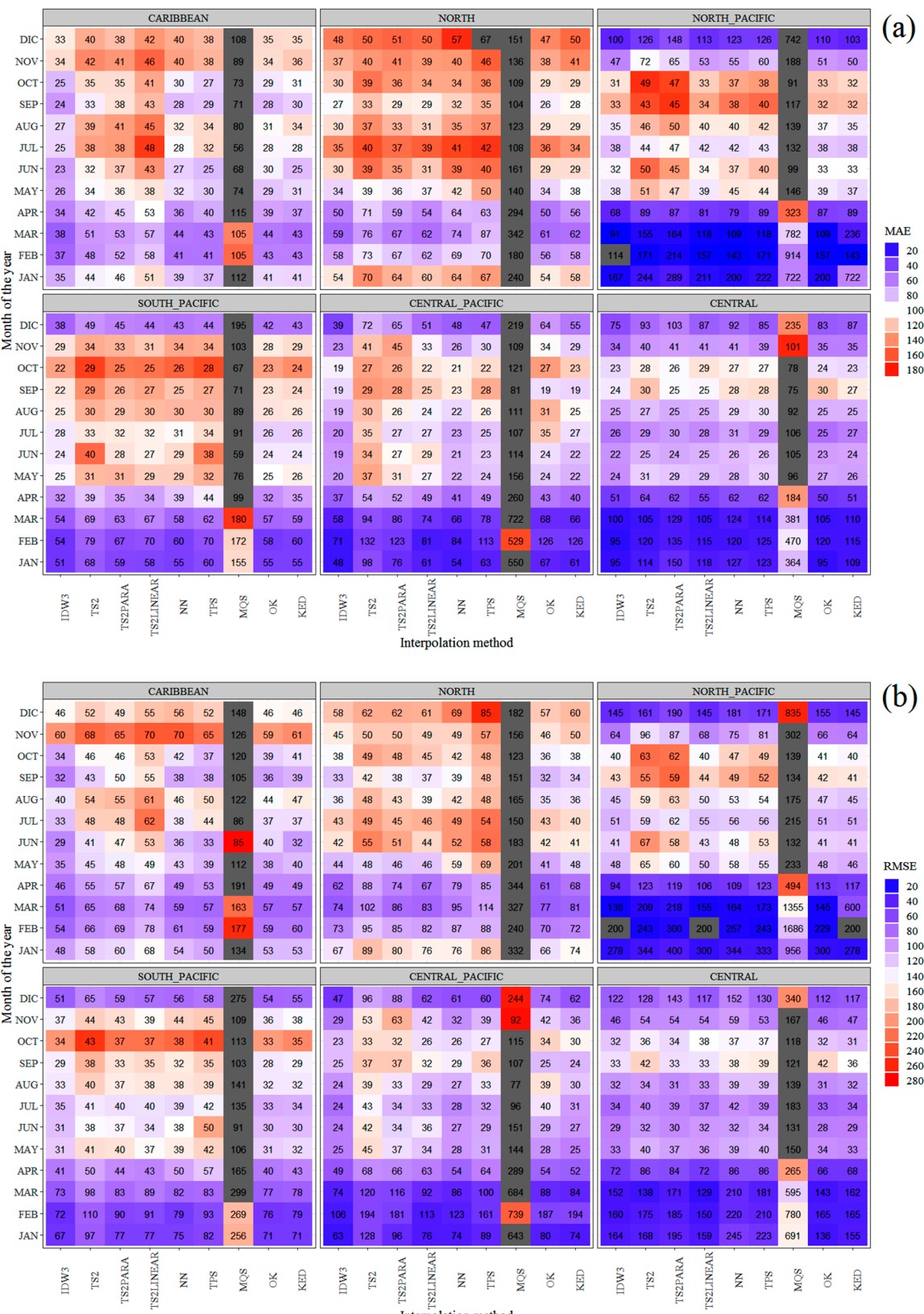

**Figure 7.** Heatmaps of monthly mean MAE (**a**) and monthly RMSE (**b**) for each climatic region during the period 1961–1990. Cell labels represent the corresponding mean objective function value as percentage of the mean monthly precipitation for the entire period.

The results also suggest that IDW is marginally superior to OK and KED regarding statistics metrics and computational efficiency. Not only is IDW relatively accurate, but it is computationally more efficient than functional minimization such as TPS and MQS or spatial covariance-based methods such as OK and KED. Several studies have found comparable performances between IDW and numerous variations of kriging interpolation [28,56–58]. Otieno et al. [28] showed that IDW and OK performed better than NN and TPS methods at various rain-gauge densities. The performances they obtained from IDW and OK were similar, suggesting that OK though complex in nature, does not show greater predictive ability than IDW.

KED did not significantly benefit from the inclusion of elevation as a covariant, since KED ranked below OK (Table 2), suggesting that in most cases the inclusion of elevation would mostly result in higher model uncertainty. This behaviour is supported by Ly et al. [57], whose results in the comparison of IDW, NN and several kriging methods showed that incorporating elevation into KED and OCK did not improve the interpolation accuracy of average daily precipitation at a catchment scale. Similarly, Dirks et al. [56] compared various kriging interpolation methods against IDW and NN in a catchment with a dense rain-gauge network, with their results showing that IDW performed slightly better.

In similar circumstances, during the calibration of the SWAT hydrological model over the Pengxi River basin of the Three Gorges Basin in China, Cheng et al. [58] found very similar results in the comparison of precipitation interpolation products generated using the Thiessen Polygon (TP), Inverse Distance Weighted (IDW) and Co-Kriging (CK) interpolation methods, and determined that IDW outperformed all methods in terms of the median absolute error.

Regardless of interpolation method, mean monthly MAE (Figure 7a) and RMSE (Figure 7b) deviations increase as monthly precipitation also increases (Figure 2), particularly for the Caribbean, North and South-Pacific regions, which suggests that precipitation temporal and spatial variability is higher during the wettest months as a consequence of orographic and convective precipitation mechanisms that are not always captured by the HSR sub-network. Higher deviations during the wettest months are even more extreme for the remaining deterministic methods, which suggest that these methods are unable to properly capturing the true-nature of spatial precipitation patterns over these regions, especially during the rainy season.

During the driest months (December to May), even when the corresponding MAE and RMSE values for the Central-Valley, North-Pacific and Central-Pacific regions are relatively low; when expressed as percentage with respect to the corresponding mean monthly precipitation, their relative importance increases considerably, demonstrating that during these months, mean monthly precipitation is not only relatively low but also highly variable (Figure 2).

The marginally lower performance of OK and KED as compared to IDW could be attributed to: (1) data stationarity and normality required by OK and KED cannot always be satisfied by Box–Cox transformation. Possible values for "$\lambda$" were constrained to a minimum value of 0.2 in order to avoid excessive data transformation, which might not be the most appropriate power for all time steps, particularly during the driest months; (2) data back-transformation may result in a biased estimation of the primary precipitation variable (mm/months). Close to logarithmic transformations introduce a positive bias in the residual distribution, which is related to an exaggeration of the upper tail of the resulting PDF. This excessive skewness ultimately could lead to overestimates in kriging precipitation estimates and variance; (3) the mechanical models selected for variogram auto-fitting (Sph, Exp, Gau, Mat and Sten) may not be sufficient to capture specific spatial structures of particular time lapses, and therefore could represent a disadvantage of the automation process. During the wettest months, convective precipitation events are extremely localized, for which the spatial variability of precipitation at a pixel scale causes the rain-gauges to disagree more profoundly among themselves. Convective storms travel in different directions and their magnitudes also vary. The distribution of precipitation changes from one event to another as well. Consequently, a wider family of mechanical variograms models should be evaluated; (4) even when a resolution of $1 \times 1$ km was chosen for the entire Costa

Rican territory, mainly on the grounds of computational costs, the impact of various spatial resolutions of the interpolation process should also be evaluated; (5) resampled elevation from the ALOS DEM was preferred over the actual rain-gauge station-elevations in order to improve spatial representation of orographic and convective precipitation mechanisms. Nonetheless, this approach could have generalized topographic information to an undesired level. KED however, performs slightly superior than OK during the wettest months for the North-Pacific and Central-Pacific regions. In contrast to the general tendency shown by IDW, OK and KED; methods NN and TPS rank respectively in the second and third positions for the Central-Pacific region. This could be related to the aforementioned high concentration of rain-gauge stations along the Pacific coastline, with only a few stations in the mountainous areas (Figure 1a). Once again, the rain-gauge network captures spatially localized precipitation events for this region throughout the year, irrespectively of dry or wet, which ultimately affects most of the geostatistical methods assumptions.

### 3.4. Comparison with Global Validation Datasets

The generated IDW climatology is generally in good agreement with GPCC, CRU and WorldClim global datasets for the Caribbean, Central-Valley, Central-Pacific, North-Pacific and North climatic regions during the period 1961–1990 (Figure 8). GPCC shows in general, the highest average correlations (0.913, 0.994, 0.940, 0.917 and 0.886, respectively) and lowest MAE (28.860, 10.759, 21.670, 13.407 and 36.903 mm/month, respectively) and RMSE (35.209, 12.786, 28.405, 15.453, and 48.959 mm/month, respectively) deviations for these five regions. The Central-Valley region shows the highest CORR and the lowest MAE and RMSE deviations, whereas the North region shows the lowest correlation and the highest MAE and RMSE deviations. This is somehow expected, as the Central-Valley has the highest observational density of all climatic regions and the North region, one of the lowest observational densities (Figure 3). Furthermore, for the period 1987–1990, the North region has only four operational rain-gauge stations, which makes the comparison between the HTR and the HSR sub-networks meaningless, since four is the minimum number of stations needed to estimate a variogram. CRU on the other hand, generally shows lower correlations (0.884, 0.926, 0.896, 0.910 and 0.884, respectively) and higher MAE (38.509, 25.103, 30.541, 14.869 and 35.567 mm/month, respectively) and RMSE (49.031, 27.736, 40.466, 17.405, 45.133 and 42.028 mm/month, respectively) deviations for these five regions when compared to GPCC, particularly for the Central-Valley region, where a considerable drop in correlation can be seem between the wettest months (September and November). A similar drop can be seen for the Caribbean region during the month of April, which produces the lowest correlation of the CRU dataset (0.884) and highest RMSE deviation (49.031 mm/month). WorldClim typically shows average correlations (0.908, 0.956, 0.927, 0.932 and 0.893, respectively), and MAE (26.591, 25.079, 26.961, 16.034 and 25.799 mm/month, respectively) and RMSE (36.871, 29.170, 37.654, 19.276 and 32.889 mm/month, respectively) deviations between GPCC and CRU, rating some months even better than GPCC. WorldClim dataset also exhibited a drop in correlation and an increase in deviation during the driest months (December to April) for the Central-Valley region, which has the highest observational density of all climatic regions (Figure 3).

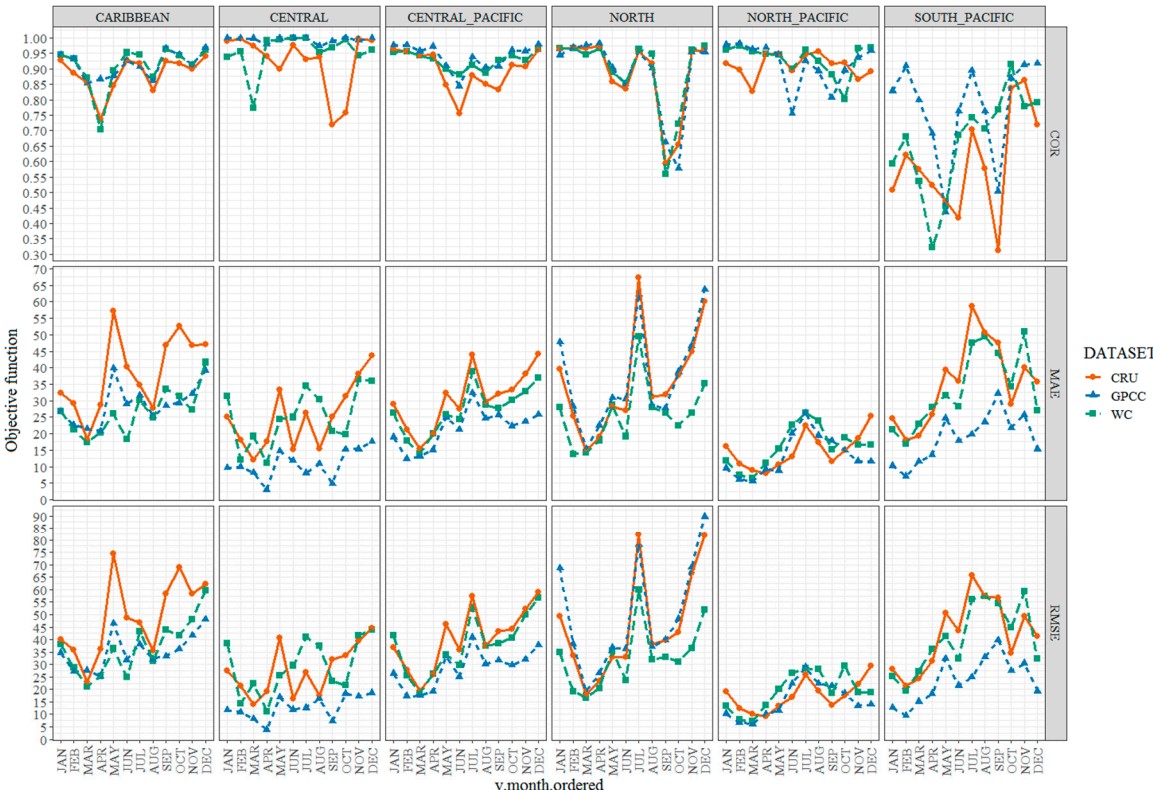

**Figure 8.** CORR, MAE and RMSE monthly performance of the generated IDW climatology against CRU, GPCC and WorldClim (WC) global datasets for each climatic region of Costa Rica during the period 1961–1990.

Regardless of the global dataset, there is an overall increasing tendency in deviation (both MAE and RMSE) for these five climatic regions as monthly precipitation moves from the dry to the wet season; with the lowest deviations found in the North-Pacific region. Furthermore, all three global datasets exhibit a considerable drop in correlation during the wettest months (August to November) for the North region, with a corresponding peak in deviation between the transitional months of June and July. The South-Pacific region however, exhibits an entirely different pattern when compared to the remaining climatic regions, since GPCC, CRU and WorldClim exhibit the lowest average CORR (0.773, 0.594 and 0.665, respectively) of all climatic regions, even when average MAE (18.607, 35.429 and 33.589 mm/month, respectively) and RMSE (23.683, 42.028 and 40.577 mm/month, respectively) are similar in proportion to other regions.

CRU and WorldClim climatologies exhibit even lower CORR values for most of the year for this climatic region. The causes of such discrepancies, which include all climatic regions but especially the South-Pacific region can generally be attributed to: (1) the number of rain-gauge stations included in the reconstruction and interpolation of the generated IDW climatology, which includes all rain-gauge stations from the HTR sub-network regardless of the duration of the recording period. The total amount of stations included in the three global dataset is most likely lower than the 416 rain-gauge stations active during the period 1961–1990; (2) the blended data from different remote-sensing products and the number of rain-gauge stations used in each global datasets. The algorithms used for estimation of precipitation from satellite radiances may underestimate precipitation, particularly for those satellites with visible and infrared sensors [9]; (3) the influence of orographic and convective precipitation mechanisms that are not properly captured by any of the methods used in the generation of each global datasets [59]; and (4) the complex topography and low rain-gauge density in the highlands and valleys, particularly in the peninsular area of the South-Pacific region (Figure 1a).



Even when the spatial resolution of the included global datasets (GPCC, CRU and WorldClim) is much lower than that of the generated IDW climatology, the overall spatial and temporal coherence among these products is considered satisfactory for all climatic regions except for the South-Pacific region. In general, however, GPCC captures more of the spatial and temporal precipitation variability within most climatic regions of Costa Rica.

This gives sufficient assurance that this new climatology can be used in the development of local climate impact studies, which are needed to provide fine-scale climate information for impact assessment and adaptation purposes. Furthermore, this new climatology is a valuable source of information to validate and statistically downscale GCM and RCM precipitation projections over the Costa Rica territory.

## 4. Conclusions

The generation of monthly precipitation climatologies for Costa Rica using irregular rain-gauge observational networks was evaluated through the application of kriging variance-reduction techniques. The following conclusions can be drawn:

i.    Based on the analysis of the Kriging Reduction efficiency objective function (KRE), the HSR sub-network captures a more detailed spatio-temporal distribution of the precipitation patterns over most climatic regions of Costa Rica even when the network is temporally discontinuous and irregularly distributed. Consequently, in the case of Costa Rica, it is better to increase the density of the rain-gauge network at the expense of temporal fidelity by including more stations even though their records may not exactly represent the same time step.

ii.   The accuracy of the spatial interpolation methods and the subsequent estimation of the KRE increases as rain-gauge sample densities also increase. Conversely, rain-gauge densities above 250 km$^2$/gauge severely impact KRE regardless of the climatic region.

iii.  IDW interpolation is marginally superior to OK and KED concerning error metrics and computational efficiency. The remaining deterministic interpolation methods considerably deviate from IDW, OK or KED, which suggests that these methods are incapable of properly capturing the true-nature of spatial precipitation patterns over the considered climatic regions.

iv.   Box–Cox data transformation is more effective in regions with lower network densities. In regions with higher network densities however, data transformation is more effecting during the wettest months. Data stationarity and normality required by OK and KED cannot always be satisfied by Box–Cox transformation, which might not be the most appropriate technique for all time steps.

v.    The mechanical models selected for variogram auto-fitting may not be sufficient to capture specific spatial structures of particular time lapses and therefore could represent a disadvantage of the adopted automation process. In spite of the fact that elevation and the orientation of the major cordilleras in Costa Rica are important modifiers of local precipitation patterns, their exact role in enhancing or reducing precipitation in the various climatic regions is a question open to debate.

vi.   The overall spatial and temporal coherence of the generated IDW climatology with GPCC, CRU and WorldClim global datasets is considered satisfactory for all climatic regions except for the South-Pacific region. Nonetheless, GPCC performed generally better in capturing the spatial and temporal patterns of observed precipitation in most climatic regions. Such agreement gives assurance about the use this new climatology in the development of local climate impact studies, which are needed to provide fine-scale climate information for impact assessment and adaptation purposes.

**Supplementary Materials:** The following are available online at http://www.mdpi.com/2073-4441/11/1/70/s1, Supplementary File 1: R-code, raw-data, results and ggplot2 graphic-code.

**Author Contributions:** M.M. and L.-A.C.-V. designed the project and drafted the manuscript; M.M. coded the entire R software; B.M. and L.-F.A.-G. provided writing ideas and supervised the study; M.M. and B.M. edited and finalized the manuscript; M.M. and L.-F.A.-G. collected and curated the observed data. All authors reviewed the manuscript.

**Funding:** This research received no external funding.

**Acknowledgments:** This research was supported by Vicerrectoría de Investigación & Extensión, Instituto Tecnológico de Costa Rica (TEC) specifically for the research project entitled "*Evaluación del impacto del Cambio Climático futuro sobre cuencas hidrológicas destinadas al abastecimiento de agua potable en Costa Rica*". The authors are grateful to D.G. Rossiter at the University of Twente, The Netherlands for his support in the conceptualization of this paper. The authors would also like to thank the editors and anonymous reviewers for their helpful and constructive comments that greatly contributed to improving the final version of this manuscript.

**Conflicts of Interest:** The authors declare no conflict of interest in any aspect of the data collection, analysis or the preparation of this paper.

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
