# Peer review of "Generation of Monthly Precipitation Climatologies for Costa Rica Using Irregular Rain-Gauge Observational Networks"

_water, doi:10.3390/w11010070_

Round 1

Reviewer 1 Report

This manuscript research on generation of monthly precipitation climatologies. It is very useful research for hydro-meteorological studies of Costa Rica. I think it can be considered for publication after a minor revision.

Specific comments:

1.     Page 1-3, Introduction section. It is too long for introduction section. You need separate it into three or four paragraphs which is easy for the reader.

2.     Why you only use the data from 1960 to 1990? Do you have some new data up to 2016?

3.     In Figure 1, I saw some stations are almost overlapping with the other stations. Is it necessary to have the similar stations for this study?

4.     Some small spelling mistake should be checked.

5.     Please try to cite the following references:

Pingping Luo, Dengrui Mu, Han Xue, Thanh Ngo-Duc, Kha Dang-Dinh, Kaoru Takara, Daniel Nover, Geoffrey Schladow, Flood inundation assessment for the Hanoi Central Area, Vietnam under historical and extreme rainfall conditions, Scientific Reports, 2018, 8:12623, DOI:10.1038/s41598-018-30024-5.

Pingping Luo, Meimei Zhou, Hongzhang Deng, Jiqiang Lyu, Wenqiang Cao, Kaoru Takara, Daniel Nover, S. Geoffrey Schladow, Impact of forest maintenance on water shortages: Hydrologic modeling and effects of climate change, Science of the Total Environment, 2018,615, pp. 1355-1363.

Pingping LUO, APIP, Bin He, Weili Duan, Kaoru Takara, and Daniel Nover: Impact assessment of rainfall scenarios and land-use change on hydrologic response using synthetic Area IDF curves, Journal of Flood Risk Management, 2018Vol.11, pp.S84–S97, DOI: 10.1111/jfr3.12164.

Pingping LUO, Bin He, Kaoru Takara, Yin E Xiong, Daniel Nover, Weili Duan, and Kensuke Fukushi, Historical Assessment of Chinese and Japanese Flood Management Policies and Implications for Managing Future Floods, Environmental Science & Policy, 2015,Vol.48, pp. 265-277, DOI: 10.1016/j.envsci.2014.12.015.

Author Response

# ////////////////////////////////////////////////////////////////////////////////////

# Reviewer No 1

# ////////////////////////////////////////////////////////////////////////////////////

This manuscript research on generation of monthly precipitation climatologies. It is very useful research for hydro-meteorological studies of Costa Rica. I think it can be considered for publication after a minor revision.

Specific comments:

1_Page 1-3, Introduction section. It is too long for introduction section. You need separate it into three or four paragraphs which is easy for the reader.

ANSWER/ Introduction has been separated into several, more organized paragraphs.

2_Why you only use the data from 1960 to 1990? Do you have some new data up to 2016?

ANSWER/ We used the 1961-1990 period as it is one of the WMO standard reference-periods for long-term Climate Change assessments. Also, it is one of the of the standard IPCC climatological periods for GCM/RCM bias correction and downscaling. We do have more data, up to present day (2015 aprox.). We will be preparing two new climatologies for the periods 1971-2000 and 1981-2010, blending Remote-Sensing products along with ground-data, but that is the subject of a different paper.

3_In Figure 1, I saw some stations are almost overlapping with the other stations. Is it necessary to have the similar stations for this study?

ANSWER/ Indeed, there are several rain-gauge stations that appear quite close to one another in Figure 1, particularly in the Central-Pacific Region, along the Pacific Coast. We did execute data screening before making the final decision on which stations we were to include and/or reject, within a 1 km2 spatial-resolution threshold. On the other hand, some stations that look overlapped do not necessary recorded during the entire 1961-1990 period, since they could have recorded data for just 5 years (or less) within the HSR sub-network. In other words, they don’t necessarily belong to the same extent of time, and most likely represent station-relocation positions.

4_Some small spelling mistake should be checked.

ANSWER/ English and spelling mistakes have been checked.

5_Please try to cite the following references:

ANSWER/ The following references have been added, thank you very much for your kind suggestion.

Luo, P., Mu, D., Xue, H., Ngo-Duc, T., Dang-Dinh, K. Takara, K., Nover, D., and Schladow, G.: Flood inundation assessment for the Hanoi Central Area, Vietnam under historical and extreme rainfall conditions, Sci. Rep.,8:12623, DOI:10.1038/s41598-018-30024-5, 2018.

Luo, P., Apip, He. B., Duan, W., Takara, K. and Nover, D.: Hydrological impact assessment of rainfall scenario, J. Flood Risk Manage., 11: S84-S97, doi:10.1111/jfr3.12164, 2018.

Thank you very much for your comments and contributions.

Reviewer 2 Report

The paper studies the precipitation in Costa Rica with ground rain gauge networks and compares against several other datasets. The study has done comprehensive analysis and comparison of all the ground stations with different interpolation methods. The twenty-year climatology is derived from several stations with reliable statistics. The Costa Rica and nearby Caribbean are an interesting area for studying precipitation, where previous studies have reported the seasonal rainfall with a bi-modal pattern dependent on specific local regions. The long-term ground data are important for studying the climatology and validate satellite observation. Overall, the work can be useful to study the local precipitation.

The study does not provide a thorough analysis of climatology. Most parts of the paper are allocated to technical details and comparison of different methods. This gives the impression of a tedious technical reports without much science.

Figure 5. In addition to Figure 5, a map of the average and variation of precipitation climatology would be useful. 

The conclusion is clutter with so many bullets. I suggest combine them, cut off some technical details and present more science.

There are award phrases in many passages. Please polish the writing and English.

Author Response

# ////////////////////////////////////////////////////////////////////////////////////

# Reviewer No 2

# ////////////////////////////////////////////////////////////////////////////////////

The paper studies the precipitation in Costa Rica with ground rain gauge networks and compares against several other datasets. The study has done comprehensive analysis and comparison of all the ground stations with different interpolation methods. The twenty-year climatology is derived from several stations with reliable statistics. The Costa Rica and nearby Caribbean are an interesting area for studying precipitation, where previous studies have reported the seasonal rainfall with a bi-modal pattern dependent on specific local regions. The long-term ground data are important for studying the climatology and validate satellite observation. Overall, the work can be useful to study the local precipitation.

Specific comments:

1_The study does not provide a thorough analysis of climatology. Most parts of the paper are allocated to technical details and comparison of different methods. This gives the impression of a tedious technical report without much science.

ANSWER/ the main objective of this study was “ to determine whether a high spatial resolution (HSR) rain-gauge network yields a more accurate estimate of average precipitation than a high temporal resolution (HTR) rain-gauge network, based on the application of kriging variance-reduction techniques”. This was possible through the implementation of the Kriging Reduction efficiency objective function (KRE), within a very complex computational framework comprised of several thousand lines of R-code, which will be openly available for researcher facing rain-gauge irregular networks like the ones we have in Costa Rica. Once we demonstrated that the HSR can be used to generate a reliable precipitation climatology, we then proceeded to test various interpolation methods and compare the results with globally available datasets.

We do acknowledge that further analysis of the generated climatology could be undertaken within specific local Climate Change studies, which by the way, we are executing at the present moment with the CORDEX Central America RCM model ensemble; but that is the subject of a different paper.

The methodology and computational components of the paper are indeed very technical, but that is the only way to justify such an effort.

2_Figure 5. In addition to Figure 5, a map of the average and variation of precipitation climatology would be useful.

ANSWER/ Could you please provide more details about the map you suggest? We also find such a map very useful, but we think that it is not directly related to Figure 5. We believe it could be part of the section “3.3. Comparison with Global Validation Datasets”, in relation to the global datasets. Would you recommend a Facet-Map of the 12 months including each Climatic Region? At 1x1 or 25x25 km spatial resolution?

3_The conclusion is clutter with so many bullets. I suggest combine them, cut off some technical details and present more science.

ANSWER/ Conclusion arguments have been combined and organized.

4_There are award phrases in many passages. Please polish the writing and English.

ANSWER/ Writing and English style have been polished and highlighted in red.

Thank you very much for your comments and contributions.